# TiBCN-Ceramic-Reinforced Ti-Based Coating by Laser Cladding: Analysis of Processing Conditions and Coating Properties

**Yuxin Li [1], Pengfei Zhang [2], Peikang Bai [1,\*], Keqiang Su [1] and Hongwen Su [3]**

[1] Shanxi Key Laboratory of Controlled Metal Solidification and Precision Manufacturing, School of Materials Science and Engineering, North University of China, Taiyuan 030051, China; liyuxin@nuc.edu.cn (Y.L.); ck603110622@163.com (K.S.)

[2] Lab. of Materials Welding and Joining, School of Materials Science and Engineering, University of Science and Technology Beijing, Beijing 100083, China; B20180166@xs.ustb.edu.cn

[3] Shanxi Yuhua Remanufacture Technology Co., Ltd., Changzhi 046000, China; sxszshw@163.com

\* Correspondence: baipeikang@nuc.edu.cn; Tel.: +86-351-355-7439; Fax: +86-351-392-2012

**Abstract:** In this paper, TiBCN-ceramic-reinforced Ti-based coating was fabricated on a Ti6Al4V substrate surface by laser cladding. The correlations between the main processing parameters and the geometrical characteristics of single clad tracks were predicted by linear regression analysis. On this basis, the microstructure, microhardness, corrosion resistance, and wear resistance of the coating and the substrate were investigated. The results showed that the clad height, clad width, clad depth, and dilution rate depended mainly on the laser power, the powder feeding rate, and the scanning speed. TiBCN-ceramic-reinforced Ti-based coating was mainly composed of directional dendritic TiBCN phases, equiaxed TiN phases, needle-like $Al_3Ti$ phases, and Ti phases. The microhardness gradually increased from the bottom to the top of the coating. The highest microhardness of coating was 1025 HV, which was three times higher than that of the Ti6Al4V substrate (350 HV). Furthermore, the coating exhibited excellent corrosion resistance and wear resistance. The corrosion potential ($E_{corr}$) reached $-1.258$ V, and the corrosion density ($I_{corr}$) was $4.035 \times 10^{-5}$ A/cm$^2$, which was one order lower than that of the Ti6Al4V substrate ($1.172 \times 10^{-4}$ A/cm$^2$). The coating wear mass loss was 4.35 mg, which was about two-third of the wear mass loss of the Ti6Al4V substrate (6.71 mg).

**Keywords:** laser cladding; geometrical characteristics; microstructure; microhardness; corrosion resistance; wear resistance

## 1. Introduction

Ti6Al4V is widely used in aerospace, petroleum, the chemical industry, and other fields due to its high specific strength and good corrosion resistance [1–3]. However, its poor wear resistance property restricts its application [4].

In order to overcome this shortcoming, researchers have paid much attention to various methods for coating the surface of this material, including the arc spraying method [5], electroplating method [6], plasma spraying method [7], and so on. The laser cladding technique is widely chosen to prepare surface coating in machined parts due to its advantages of higher mechanical properties and lower metallurgic defects [8]. The effect of processing parameters (laser power ($P$), scanning speed ($S$), powder feeding rate ($F$), and so on) on the clad geometry, microstructure, and properties of coatings has been reported [9–15]. Sun et al. [16] studied the influence of laser processing parameters on the geometrical characteristics of Ti6Al4V coating. The results indicated that the powder feed rate was the dominant factor on the clad width and clad height, while laser scanning speed had the strongest effect on clad

depth. Lin et al. [17] studied the effect of processing parameters on the properties of TiB$_2$/TiB cladding coatings on Ti6Al4V alloy. The results indicated that the microhardness and wear resistance of the coatings gradually increased with increasing laser power density. Li et al. [18] fabricated laser cladding coatings on a Ti6Al4V substrate, and the results showed that the coatings had better wear resistance. Kumar et al. [19] studied the characterization of clad layer morphology, microhardness, wear behavior, and elemental compound formations of AlN-Ni-Ti6Al4V coating. The results found that the cladding microhardness increased up to 3–4 times that of the substrate. Furthermore, the coating had better wear resistance. Based on the above analyses, it can be seen that researchers have added and synthesized binary ceramic materials (such as TiC, TiB, TiB$_2$, and AlN particles) to increase the properties of the coating on Ti6Al4V alloy substrates. However, investigations into ternary and quaternary ceramic composite coatings on Ti6Al4V alloy substrates are lacking.

TiBCN is a quaternary ceramic material with good mechanical properties and chemical stability [20–24]. However, there are few reports about the application of TiBCN coating in laser cladding.

In this paper, the laser cladding parameters of 90 wt.% Ti + 10 wt.% TiBCN powder were optimized by the regression analysis method. TiBCN-ceramic-reinforced Ti-based coating was fabricated on a Ti6Al4V substrate surface by laser cladding, and the microstructure and properties were investigated.

## 2. Experimental

Ti6Al4V alloys (20 mm × 15 mm × 10 mm) (YeBo Steel Co., LTD. Dongguan, China) was used as the substrate. The chemical composition of the alloy is listed in Table 1. In order to remove the oxide film, the end face of the substrate was polished with metallographic sand paper (180#) and washed with acetone before laser cladding. For the clad material, 90 wt.% Ti (99.5%, 115–230 μm, Global Golden Ding Technology Co. LTD, Beijing, China) and 10 wt.% TiBCN (99.5%, 115–230 μm) were used.

**Table 1.** Chemical composition (wt.%) of Ti6Al4V alloy.

| Element | Content |
|---------|---------|
| Al | 6.01 |
| V | 3.84 |
| Fe | 0.3 |
| S | 0.15 |
| C | 0.1 |
| O | 0.15 |
| N | 0.15 |
| Ti | Bal. |

A 4400 W laser system (LDF 4000-100, Laserline Gmbh, Mülheim-Kärlich, Germany) was used to produce the coatings. The coaxial powder delivery system (DMS-3D, Duomu Industry Co., Ltd., Shanghai, China) was used to feed powders, and high-purity argon was used as the laser shielding gas. The main parameters of laser cladding processing were chosen as follows: laser power $P$ = 800–1400 W, scanning speed $S$ = 3–7 mm/s, powder feeding rate $F$ = 200–300 mg/s, overlap rate $D$ = 50%.

The clad height ($h$), clad width ($w$), and clad depth ($b$) of the tracks was measured using the MIAPS software (Release version 5.7, Precise Instrument Co., Ltd., Beijing, China). Figure 1 shows the schematic view of a typical single cladding. The dilution ($D$) was calculated using Equation (1) [16]:

$$D(\%) = \frac{b}{b+h} \qquad (1)$$

For predicting the different laser track geometry characteristics, the model was established by Equation (2) [25]:

$$y = a(P^{\alpha}S^{\beta}F^{\gamma}) + b \qquad (2)$$

where $y$ is the measured geometrical characteristics values; a and b are constants.

Taking the logarithm of both sides, Equation (3) was derived:

$$\ln y = \ln a + \alpha \ln P + \beta \ln S + \gamma \ln F + \ln b \tag{3}$$

If $y_1 = \ln y$, $X = \ln P$, $Y = \ln S$, $Z = \ln F$, $A = \ln a + \ln b$, Equation (4) would be as follows:

$$y_1 = \alpha X + \beta Y + \gamma Z + A \tag{4}$$

$\alpha$, $\beta$, $\gamma$ were used to determine the linear regression analysis, which were calculated using SPSS Statistics analysis software (SPSS, Statistics 19).

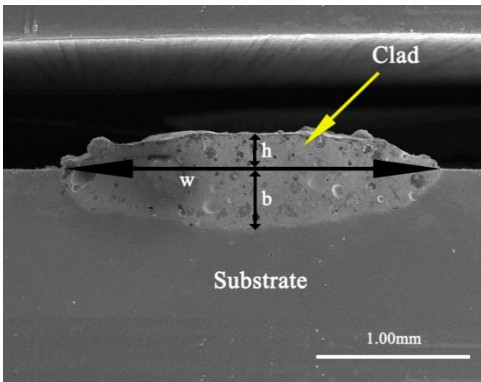

**Figure 1.** Schematic representation of geometrical characteristics in a single cladding.

D/Max 2500PC Rigaku X-ray diffractometer (Spring, TX, USA) was used to detect phase identification of the coatings using Cu-K$\alpha$ radiation. SEM (INSPECTF50, FEI, Hillsboro, OR, USA) equipped EDS was used for the morphology of the coating. High-resolution transmission electron microscope (HRTEM, JEM-2100F, JEOL Ltd., Tokyo, Japan) was also employed to identify the main phases of the coating. The TEM samples were mechanically polished to a thickness of about 50 μm and then thinned by a twin-jet electropolisher using a solution of 10% perchloric acid and 90% methanol (vol.%)/HClO$_4$:C$_2$H$_4$O$_2$ = 1:9 (vol.) under a voltage of 30 V at room temperature. The microhardness of the coatings was tested using a JMHVS-1000AT Viskers hardness tester (Shanghai Aolong Xingdi Testing Equipment Co., LTD, Shanghai, China). The polarization curves of the coating and substrate were acquired by CHI660E with the specimen as the working electrode, a graphite plate as the auxiliary electrode, and a saturated calomel electrode as the reference electrode. The test medium was a 3.5 wt.% NaCl solution, the water bath kettle temperature was 25 °C, and the scanning rate was 1 mV/s. The sliding wear test was performed by HSR-2M (Lanzhou Zhongke Kaihua Technology Development Co., Ltd, Lanzhou, China) at 25 °C without lubrication.

## 3. Results and Discussion

### 3.1. Effect of Different Processing Parameters

Figure 2 shows the effect of $P$, $S$, and $F$ on the clad height ($h$). It can be seen that the clad height increased when the powder feeding rate increased and the scanning speed decreased. Furthermore, it can also be seen that the clad heights were approximately equal with different values of laser power when the powder feeding rate and scanning speed had the same values, indicating that the laser power is a negligible parameter for predicting the clad height. Figure 3 shows the relationship between the clad height and the two processing parameters ($S$ and $F$). The results show that the clad height is a function of $S^{-3/4}F$ with a linear regression coefficient $R^2 = 0.96$. The high $R^2$ confirms that the statistical

linear model is valid. Hence, the clad height is just controlled by *S* and *F*, which is in agreement with previous works [25–27].

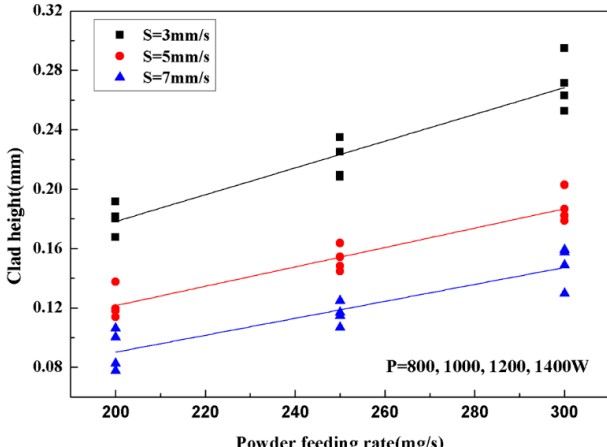

**Figure 2.** The effect of the main processing parameter on the clad height (*h*).

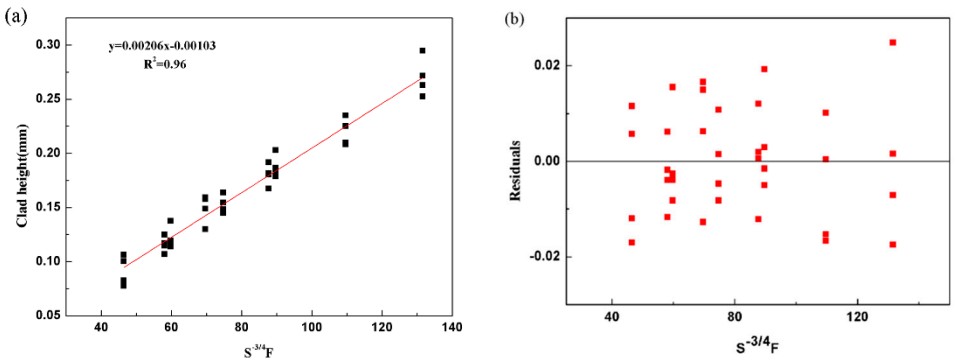

**Figure 3.** (**a**) The relationship between clad height and the two processing parameters (scanning speed (*S*) and powder feeding rate (*F*)). (**b**) Residuals.

Figure 4 shows the effect of *P*, *S*, and *F* on the clad width (*w*). The results show that the clad width increased when the laser power increased and the scanning speed decreased. The clad widths were approximately equal when the laser power and the scanning speed were the same values and the powder feeding rates were different values. This indicates that the scanning speed and the laser power are two important parameters, but the powder feeding rate is a dispensable parameter. Figure 5 shows the relationship between the clad width and the three processing parameters (*P*, *S*, and *F*). It can be seen that the clad width is controlled by the $P^{2/5}S^{-1/2}F^{1/5}$ parameter with the linear regression coefficient $R^2 = 0.98$. The high $R^2$ confirms that the statistical linear model is valid.

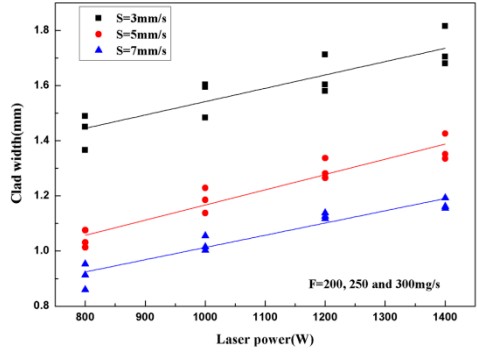

**Figure 4.** The effect of the main processing parameter on the clad width (*w*).

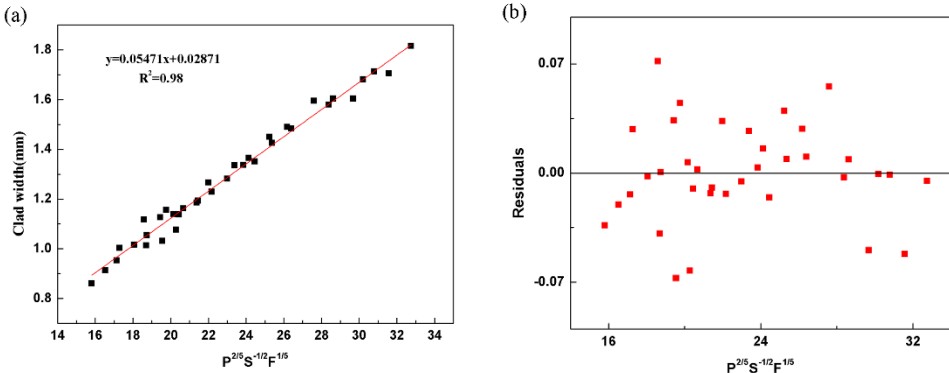

**Figure 5.** (**a**) The relationship between clad width and the three processing parameters (laser power (*P*), *S*, and *F*). (**b**) Residuals.

Figure 6 shows the effect of *P*, *S*, and *F* on the clad depth (*b*). The results show that the laser power and the scanning speed have an important effect on the clad depth. Furthermore, there was a strong linear dependence of clad depth on the laser power when the scanning speed and the powder feeding rate were all of the same values. Figure 7 gives the relationship between the clad depth and the two processing parameters (*P* and *S*). It can be seen that the clad depth is controlled by $P^{4/5}S^{3/5}$ with $R^2 = 0.94$. Hence, the clad depth is just controlled by laser power (*P*) and scanning speed (*S*).

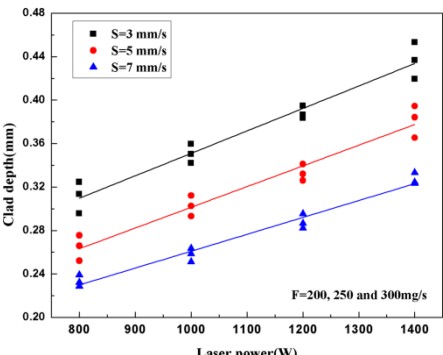

**Figure 6.** The effect of the main processing parameter on the clad depth (*b*).

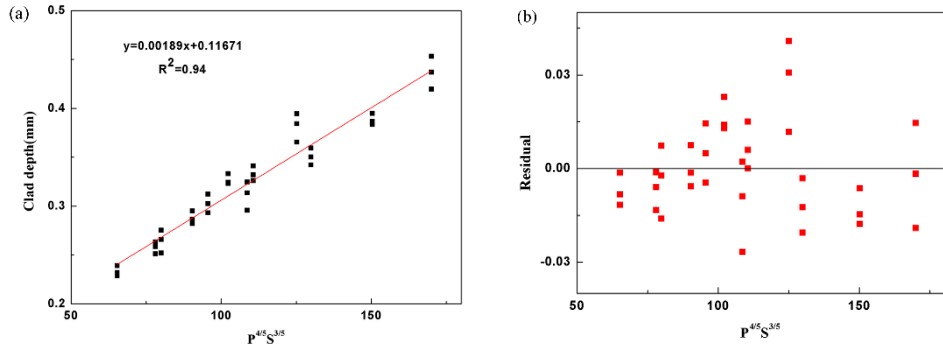

**Figure 7.** (**a**) The relationship between clad depth and the two processing parameters (*P* and *S*). (**b**) Residuals.

Figure 8 gives the effect of *P*, *S*, and *F* on the dilution rate (*D*). It can be seen that the effect of scanning speed on the dilution rate was opposite to that of powder feeding rate. Figure 9 shows the relationship between the dilution rate and *P*, *S*, and *F*. It can be observed that the dilution rate is a function of $P^{1/2}S^{-4/5}F^{1/10}$ with regression $R^2 = 0.88$. Hence, the dilution rate is controlled by *P*, *S*, and *F*.

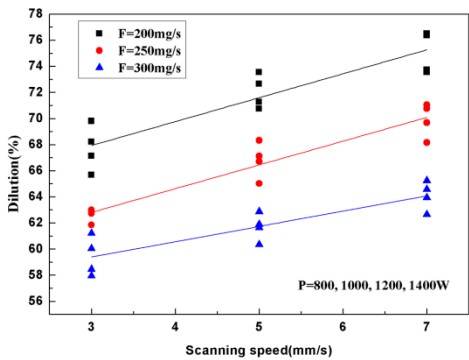

**Figure 8.** The effect of the main processing parameter on the dilution rate.

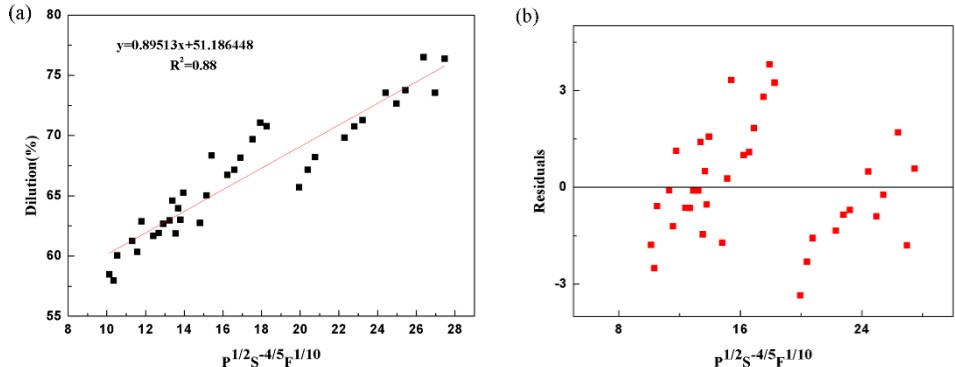

**Figure 9.** (**a**) The relationship between the dilution rate and the three processing parameters (*P*, *S*, and *F*). (**b**) Residuals.

The predicted combined parameters are shown in Table 2. It can be seen that the scanning speed, powder feeding rate and laser power have an important effect on the clad width, clad height, clad depth, and dilution of single clad tracks. According to Figure 3, Figure 5, Figure 7, and Figure 9, the processing parameter zone of laser cladding can be determined. The process parameters were selected as follows in the present study: laser power of 1200 W, powder feeding rate of 250 mg/min, and scanning speed of 5 mm/s.

**Table 2.** Predicted combined parameters representing the best correlation with measured geometrical characteristics of single cladding.

| Quantity | Combined Parameter (*x*) | $R^2$ | a | b |
|---|---|---|---|---|
| *h* (mm) | $S^{-3/4}F$ | 0.96 | 0.00206 | −0.00103 |
| *w* (mm) | $P^{2/5}S^{-1/2}F^{1/5}$ | 0.98 | 0.05471 | 0.02871 |
| *b* (mm) | $P^{4/5}S^{3/5}$ | 0.94 | 0.00189 | 0.11674 |
| *D* (mm) | $P^{1/2}S^{-4/5}F^{1/10}$ | 0.88 | 0.89513 | 51.186448 |

### 3.2. Phase Composition and Microstructure

Figure 10 shows the XRD pattern of TiBCN-ceramic-reinforced Ti-based coating. It can be noted that the TiBCN-ceramic-reinforced Ti-based coating contained Ti, TiBCN, TiN, and $Al_3Ti$ phases. The presence of the TiN phase indicates that TiBCN decomposition occurred during laser cladding.

Figure 11 shows SEM image of transverse cross-section of TiBCN-ceramic-reinforced Ti-based coating. Figure 11a shows the cross-section SEM morphology at low magnification. It can be seen that there was a clear fusion line between the substrate and the coatings, which proves that there was good metallurgical fusion between the two. Figure 11b clearly shows the different microstructural morphologies of the interface region between the coating and the substrate. It can be found that the

equiaxed grains and needle-like phases appeared in the interface region. Table 3 gives EDS analysis results of the clad coating. Combined with the EDS and XRD results (see Figure 10), it can be seen that the equiaxed grains (Point A) are TiN phases, and the needle-like phases (Point B) are $Al_3Ti$ phases. Figure 11c shows the microstructural morphology of the upper part of the cladding coating. It can be seen that the microstructure was mainly composed of directional dendritic phases at the upper part of the clad coating. Figure 11d is a magnification of the square portion of Figure 11c. According to Table 3, the dendritic phases (Point C) are TiBCN phases.

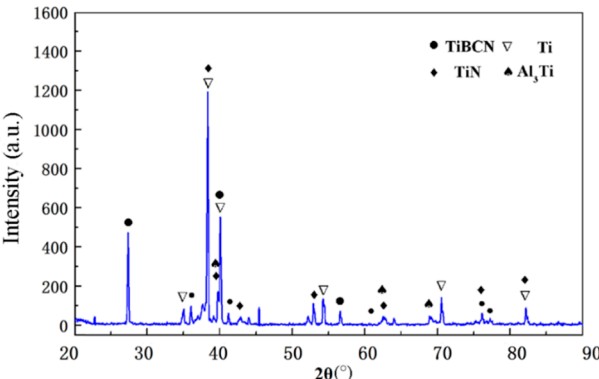

**Figure 10.** The XRD pattern of Ti/TiBCN coating.

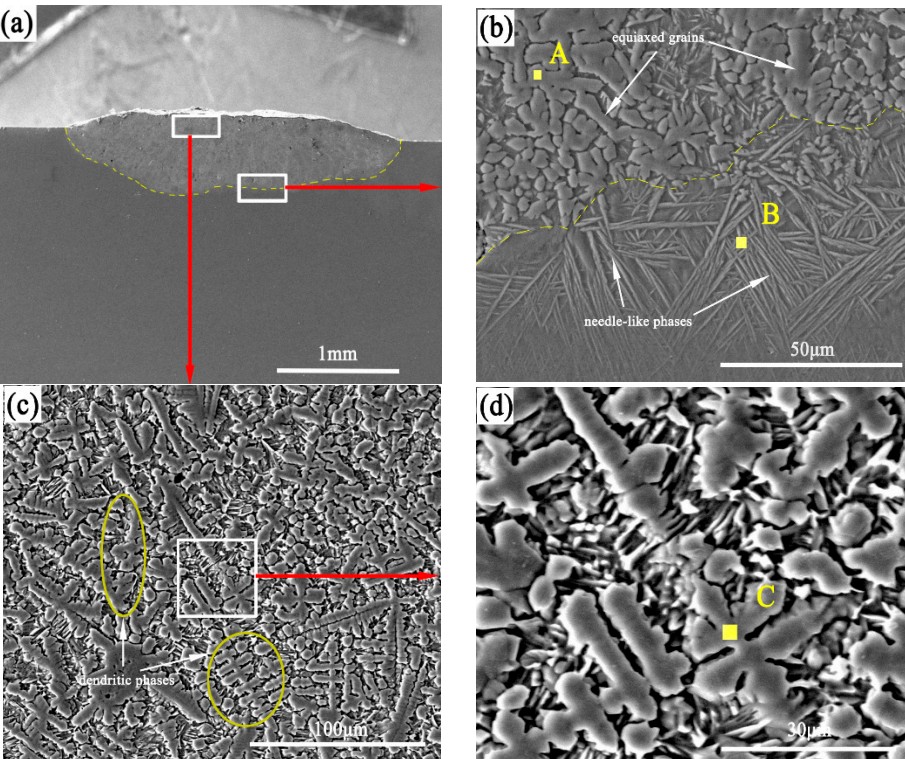

**Figure 11.** SEM image of transverse cross-section of TiBCN-ceramic-reinforced Ti-based coating. (**a**) Cross-sectional macromorphology of the coating; (**b**) the interface region; (**c**) top of coating; and (**d**) the magnified image of the quadrangle in (**c**).

Further analysis was carried out on the reinforced phases of the clad coating. Figure 12 gives the bright-field HRTEM images and the corresponding selected area diffraction patterns (SADP) of TiN. Because the sample for TEM analysis was so small (Ø = 3 mm), intermetallic compounds ($Al_3Ti$) were not detected. According to the SADP analysis, the selected area electron diffraction index calibration of the TiN [011] crystal zone axis could also be located in the corresponding bright-field image. Figure 12a

shows the equiaxed grains TiN. The TiN phases had the face-centered cubic structure (FCC). As shown in Figure 12b, the TiN phase grew along (3 $\bar{3}$ 3), ($\bar{4}$ $\bar{2}$ 2), and ($\bar{1}$ $\bar{5}$ 5) planes.

**Table 3.** Energy-dispersive spectroscopy (EDS) analyses (in at.%) of the typical position in Figure 11.

| Point | Element (wt.%) | | | | |
|-------|------|-------|-------|-------|-------|
| | **Ti** | **B** | **C** | **N** | **Al** |
| A | 36.21 | – | 10.25 | 39.14 | 14.4 |
| B | 24.54 | – | 8.25 | 4.74 | 62.47 |
| C | 24.25 | 28.61 | 21.23 | 20.55 | 5.36 |

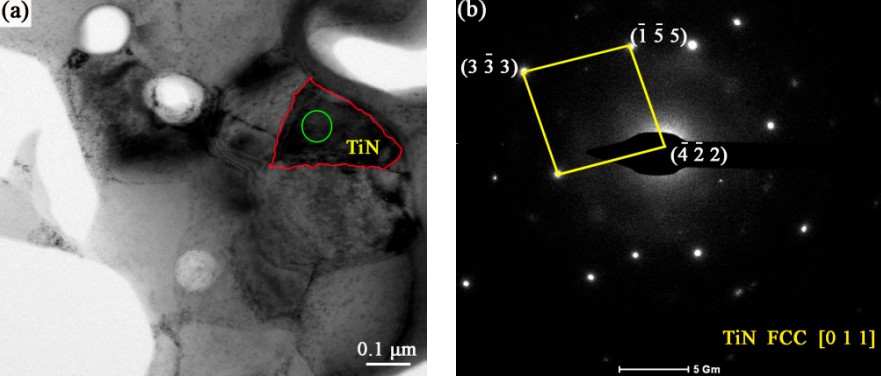

**Figure 12.** Bright-field TEM images of (**a**) reinforced particles and (**b**) selected area electron diffraction patterns.

### 3.3. Properties

Figure 13 shows the microhardness profile from the top of the coating to the substrate. The results show that the microhardness gradually decreased from the top of the coating to the substrate. The highest microhardness of the coating was 1025 HV, which was three times higher than that of the Ti6Al4V substrate (350 HV). The enhancement of microhardness of the coating was mainly due to the presence of hard and brittle TiBCN and TiN phases in the microstructure. TiN and TiBCN are interstitial compounds, and the N and C atoms in the solid solution can cause lattice distortion [28], which increases the resistance of the dislocation movement and prevents lattice slip.

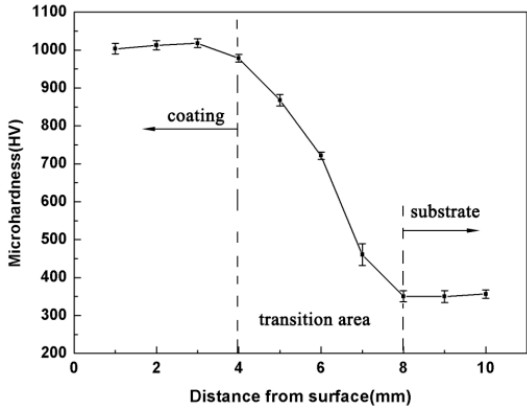

**Figure 13.** The microhardness profile from the top of the coating to the substrate.

Figure 14 gives the polarization curves of the substrate and the coating. In comparison to the Ti6Al4V substrate, the coating had higher corrosion potential ($E_{corr}$) of −1.315 V and lower corrosion density ($I_{corr}$) of 7.568 × $10^{-5}$ A/cm$^2$. Furthermore, the polarization resistance ($R_p$) of the coating

was bigger than that of the substrate, indicating that TiBCN-ceramic-reinforced Ti-based coating can improve the corrosion resistance of the Ti6Al4V alloy.

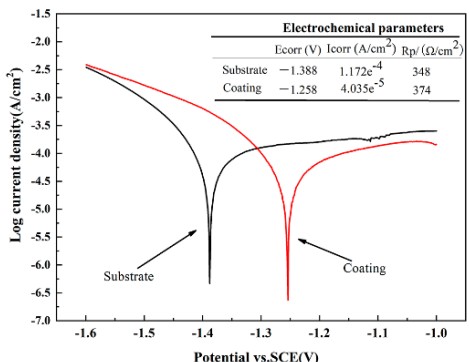

**Figure 14.** The potentiodynamic polarization curves for the coating and substrate.

Figure 15 gives the wear mass loss of the coating and the substrate. As can be found obviously, the mass loss of the coating and the Ti6Al4V substrate was 4.35 mg and 6.71 mg, respectively. Figure 16 gives the morphology of the coating and the substrate after sliding wear. It can be seen that the worn degrees of the coating were significantly less than those of the Ti6Al4V substrate. Moreover, the worn surfaces of the substrate and the coating exhibited various worn characteristics. The Ti6Al4V substrate surface displayed a lot of deep grooves and spallings. In contrast, the coating surface was relatively smooth with only some shallow and narrow grooves and no plastic deformation, indicating that TiBCN-ceramic-reinforced Ti-based coating can improve the wear resistance of the Ti6Al4V alloy.

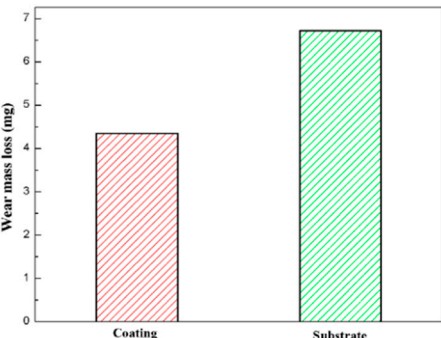

**Figure 15.** The wear mass loss of the composite coating and substrate.

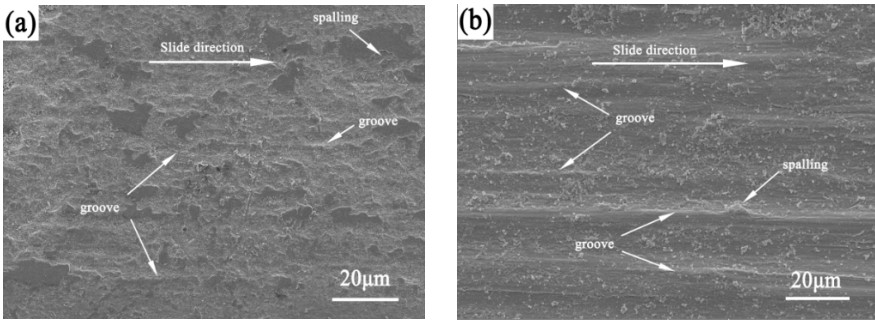

**Figure 16.** The worn surface morphologies of the composite coating and substrate: (**a**) composite coating; (**b**) substrate.

## 4. Conclusions

TiBCN-ceramic-reinforced Ti-based coating was prepared on the Ti6Al4V substrate surface, and the geometrical characteristics, microstructure and properties of the coatings were investigated. The conclusions are as follows:

- The geometrical characteristics ($h$, $w$, $b$, $D$) depend mainly on the laser power, the feeding rate, and the scanning speed. This relationship can be written in the form $S^{-3/4}F$, $P^{2/5}S^{-1/2}F^{1/5}$, $P^{4/5}S^{3/5}$, and $P^{1/2}S^{-4/5}F^{1/10}$ with a correlation coefficient $R^2$ = 0.96, 0.98, 0.94, and 0.88, respectively.
- The TiBCN-ceramic-reinforced Ti-based coating consists of the directional dendritic TiBCN phases, the equiaxed TiN phases, needle-like $Al_3Ti$ phases, and Ti phases.
- In comparison to the Ti6Al4V substrate, the microhardness, corrosion resistance, and wear resistance of TiBCN-ceramic-reinforced Ti-based coating are obviously improved.

**Author Contributions:** Data Curation, P.Z.; Investigation, K.S. and H.S.; Writing—Original Draft, Y.L.; Writing—Review & Editing, P.B.

**Funding:** This work was supported by the Natural Science Foundation of China (No. U1810112, No. 51604246) and the Taiyuan Science and Technology Project (No. 170205).

**Conflicts of Interest:** The authors declare no conflict of interest.

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
