# Peer review of "TiBCN-Ceramic-Reinforced Ti-Based Coating by Laser Cladding: Analysis of Processing Conditions and Coating Properties"

_coatings, doi:10.3390/coatings9060407_

Reviewer 1 Report

The paper deals with laser cladding for producing a TiBCN ceramic reinforced Ti based coating on Ti6Al4V. The paper is well written and structured. The authors used many techniques to evaluate their material. The paper deserves to be published. However some remarks must be clarified.

Minor remarks

1) line 15: correct the word results.

2) line 60-65: the size of the lettering is bigger

Main remarks

1) How did the authors derived with eduation 2? This part is not clear and need futher explanation. Did this equation derived from the experimental results depicted in figures 2-9?

2) The  optimal process parameters are selected by the SPSS study. Did these parameters actually gave the optimum results with respect to microstructure and properties? Is the obtained microstructure the desired one?

3) The corrosion results need more explanation. From fig 14 it seems the corrosion tendancy if the same for both materials.  In order to evaluate the corrosion protection derived by the coating,  it is better to conduct an EIS study (electrochemical impendance spectroscopy)

4) Corrosion testing parameters needs to be added in the experimental section. Did the author used a three electrode cell? Which electorde they used? etch.

5) The authors should enhace the innovative character of the paper. They should provide evidance that their coating exhibits better properties. Nevertheless, they are hardly any references in the results section indicating that! 

Author Response

Dear prof.,

  Thank you very much for your kind suggestions about our manuscript entitled “TiBCN ceramic reinforced Ti based coating by laser cladding: Analysis of processing conditions and coating properties” We have revised the whole manuscript carefully that includes the quality of the written English, accurate illustrations. All the corrections have already been made, and the changes have been highlighted in the marked up manuscript. The attachment is my point-by-point response.

Reviewer 2 Report

In this paper, The possibility of fabricating the TiBCN ceramic reinforced Ti based coating on Ti6Al4V substrate surface by laser cladding was studied. The the laser cladding parameters of 90 wt. % Ti+10 wt. %TiBCN powder were optimized by regression analysis method. In addition, the the microstructure, microhardness, corrosion resistance and wear resistance of the coating were investigated. The research is interesting and is good presented. However, to be accepted for publication the following comments need to be addressed.

1-    The proficiency of the language needs a minor improvement in the manuscript (check L 62-63) (some but not all).

2-    A lot of punctuation is missing;

3-     The Font size should be unified (Experimental) part specialized word should be unified. Such as "Figure" and "Fig" (some but not all).

4-    The introduction section needs to be improved by citing related articles of the current journal.

5-    In abstract section, L238, I would suggest writing how much these properties are improved.

6-    Subscript writing should be checked such as in line 49 and 237 (some but not all).

7-    Figure 10, please check the Y-axis.

8-    In experimental section, I would suggest explaining the preparing method of the TEM samples.

Author Response

(The authors gave the same response as above.)

Reviewer 3 Report

This is an interesting piece of work about improving the surface conditions of Ti6Al4V components by depositing TIBCN reinforced Ti-based coatings via Laser cladding. Most of the aspects about this article have been already taken care of by the Authors with only few but really important aspects ignored. Addressing some of such aspects will help in strengthening this article even more. 

English must be improved. There are some places where the sentence formulation, grammar etc. is poor.  (e.g. line 36-37, 50-51 etc.)

Major comment about this article is that Authors have not really written the experimental section in more detail. for instance:

(a)  Line 63-64: Please provide the manufcaturer details of both the Ti and TiBCN powder (Company name, City, Country).  Also comment if they are commercially available powders or if they tailored made for this research.

(c) Line 75-76: MIPAS: Please provide Manufacturer details (Comapny name, city and country).

(d) In fact this is a general comment for all the experimental equipment/materials used by authors: Add their manufacturer details if they are commercial products. If not commercial then provide more details.  

Lin 100: Define P, S and F abbreviations somewhere in the experimental sections so they don't come as surprise to the readers. 

Two major comments on the whole regression analysis:

(a) Authors should clearly explain the basis of selecting the optimized parameters. Also a more elaborate explanation is needed to help the readers understand about how authors reached to the conclusion of selecting these optimized parameters. As if now authors have just said 'based on the above-mentioned considerations (line 151)' but this can be made more obvious for the sake of readers by briefly showing the procedure used to arrive at the optimized parameters. 

(b) The second comment is about error analysis of the model. Please add a standard error analysis of the experimental results. Also mention at what confidence interval the analysis is carried out. 

(c) In the experimental section authors should also explain about what experimental design they have considered ? and  how many replicates for each experimental set/data point were used?

Author Response

Dear prof.,

  Thank you very much for your kind suggestions about our manuscript entitled “TiBCN ceramic reinforced Ti based coating by laser cladding: Analysis of processing conditions and coating properties” We have revised the whole manuscript carefully that includes the quality of the written English, accurate illustrations. All the corrections have already been made, and the changes have been highlighted in the marked up manuscript. The attachment is my point-by-point response.

Round  2

Reviewer 3 Report

Its fine to publish this work in its current form.